# Examining the dynamics of Epstein-Barr virus shedding in the tonsils and the impact of HIV-1 coinfection on daily saliva viral loads

Catherine M. Byrne[1,2,3]*, Christine Johnston[4,5], Jackson Orem[5,6], Fred Okuku[6‡], Meei-Li Huang[5,7], Habibur Rahman[8], Anna Wald[4,5,7,9], Lawrence Corey[4,5,7], Joshua T. Schiffer[4,5,10], Corey Casper[4,5,11], Daniel Coombs[2,12☯], Soren Gantt[13☯]

1 Department of Microbiology and Immunology, University of British Columbia, Vancouver, British Columbia, Canada, 2 Institute of Applied Mathematics, University of British Columbia, Vancouver, British Columbia, Canada, 3 British Columbia Children's Hospital Research Institute, Vancouver, British Columbia, Canada, 4 Department of Medicine, University of Washington, Seattle, Washington, United States of America, 5 Vaccine and Infectious Disease Division, Fred Hutchinson Cancer Research Center, Seattle, Washington, United States of America, 6 Uganda Cancer Institute, Kampala, Uganda, 7 Department of Laboratory Medicine, University of Washington, Seattle, Washington, United States of America, 8 Faculty of Medicine, University of British Columbia, Vancouver, British Columbia, Canada, 9 Department of Epidemiology, University of Washington, Seattle, Washington, United States of America, 10 Clinical Research Division, Fred Hutchinson Cancer Research Center, Seattle, Washington, United States of America, 11 Infectious Disease Research Institute, Seattle, Washington, United States of America, 12 Department of Mathematics, University of British Columbia, Vancouver, British Columbia, Canada, 13 Département de Microbiologie, Infectiologie et Immunologie, Université de Montréal, Montréal, Québec, Canada

☯ These authors contributed equally to this work.
‡ Unavailable.
* cbyrne@bcchr.ubc.ca

**Data Availability Statement:** All data files are available from the Dryad data repository (doi:10.5061/dryad.w6m905qkh).

## Abstract

Epstein-Barr virus (EBV) is transmitted by saliva and is a major cause of cancer, particularly in people living with HIV/AIDS. Here, we describe the frequency and quantity of EBV detection in the saliva of Ugandan adults with and without HIV-1 infection and use these data to develop a novel mathematical model of EBV infection in the tonsils. Eligible cohort participants were not taking antiviral medications, and those with HIV-1 infection had a CD4 count >200 cells/mm³. Over a 4-week period, participants provided daily oral swabs that we analysed for the presence and quantity of EBV. Compared with HIV-1 uninfected participants, HIV-1 coinfected participants had an increased risk of EBV detection in their saliva (IRR = 1.27, 95% CI = 1.10–1.47) and higher viral loads in positive samples. We used these data to develop a stochastic, mechanistic mathematical model that describes the dynamics of EBV, infected cells, and immune response within the tonsillar epithelium to analyse potential factors that may cause EBV infection to be more severe in HIV-1 coinfected participants. The model, fit using Approximate Bayesian Computation, showed high fidelity to daily oral shedding data and matched key summary statistics. When evaluating how model parameters differed among participants with and without HIV-1 coinfection, results suggest HIV-1 coinfected individuals have higher rates of B cell reactivation, which can seed new infection in the tonsils and lower rates of an EBV-specific immune response. Subsequently, both these traits may explain higher and more frequent EBV detection in the saliva of HIV-1 coinfected individuals.

**Funding:** This work was supported by the Natural Sciences and Engineering Research Council of Canada (Postgraduate Scholarship-Doctoral Program to CMB and Discovery grant RGPIN-2015-03611 to DC, http://www.nserc-crsng.gc.ca/), Emory University (sabbatical funding from the Quantitative Theory and Methods Initiative to DC, http://quantitative.emory.edu/about/index.html) the Canadian Institutes of Health Research (Operating Grant to SG, http://www.cihr-irsc.gc.ca/e/193.html), the Fred Hutchinson Cancer Research Center (Joel Meyers Infectious Disease Scholarship Grant to CJ and the Early Detection Initiative to LC, https://www.fredhutch.org/en.html), the Doris Duke Charitable Foundation (Clinical Scientist Development Award to CC, https://www.ddcf.org/) and the National Institutes of Health (P30 CA015704 to CC, K23 AI54162 to CC, K23 AI079394 to CC, P30 AI027757 to CC, K24 AI071113 to AW, and PO1 AI030731 to LC and AW, https://www.nih.gov/). This work was enabled in part by support provided by WestGrid and Compute Canada. The funders had no role in study design, data collection and analysis, decision to publish, or preparation of the manuscript.

**Competing interests:** The authors have declared that no competing interests exist. Author Fred Okuku was unable to confirm his authorship contributions. On his behalf, the corresponding author has reported their contributions to the best of their knowledge.

## Author summary

Epstein-Barr virus (EBV) is a ubiquitous infection worldwide associated with the development of several kinds of cancer, including B cell lymphoma and nasopharyngeal carcinoma. Rates of EBV replication and disease are higher in individuals who are coinfected with HIV-1. HIV-1 infection is associated with increased B cell activation as well as immunodeficiency resulting from loss of T cells; however, whether these factors contribute to higher rates of EBV replication during coinfection, and by how much, has remained unknown. We analysed oral EBV shedding data from a cohort of Ugandan adults taken at multiple time points and found that participants coinfected with HIV-1 maintained higher quantities of EBV in their saliva. To better understand this finding, we developed a mathematical model to describe the dynamics of EBV infection within the tonsils. By rigorously matching our model to our participant data, we determined that both high rates of infected B cell activation and worse cellular immune control of EBV may cause higher EBV loads in saliva during HIV-1 coinfection. These results help explain the impact of HIV-1 on EBV and suggest potential therapeutic targets to prevent EBV-related malignancy in people who are coinfected with HIV-1.

## Introduction

Epstein-Barr virus (EBV) infection is associated with the development of approximately 200,000 malignancies per year worldwide, including B cell lymphomas and nasopharyngeal carcinoma [1]. The risk of EBV-associated malignancies is significantly higher in people coinfected with HIV-1. For example, the risk of non-Hodgkin lymphoma, an AIDS-defining cancer, in the U.S. is 10-fold higher among HIV-1 coinfected individuals than in the general population [2]. Individuals with EBV/HIV-1 coinfection tend to have higher EBV viral loads in saliva and blood [3–5]. Uncovering the mechanisms by which HIV-1 may impair the control of EBV infection could provide clues relevant to the prevention of EBV-related disease as well as insights into basic EBV pathobiology.

EBV infection is primarily transmitted via saliva and is nearly universal, acquired during early childhood in developing countries and before reaching young adulthood in developed countries [6–8]. During primary infection, EBV is thought first to infect oral epithelial cells overlying the lymphoid tissue known as Waldeyer's ring [9]. This area consists of the tubal, adenoid, palatine and lingual tonsils [9]. Infected epithelial cells produce large numbers of infectious virions [10], facilitating latent infection of naïve B cells in the underlying lymphoid tissue. EBV drives these naïve B cells to mature into resting memory B cells and circulate throughout the body through the expression of only a small number of latent gene products [11, 12]. Viral shedding is highest during primary EBV infection but remains frequent throughout chronic infection [5]. During chronic infection, B cells latently infected with EBV can return to Waldeyer's ring, encounter cognate antigen, and become activated to mature into plasma cells, triggering lytic reactivation and production of infectious virions [13–15]. This process initiates a new round of epithelial infection in the tonsils and viral shedding in the saliva.

The dynamics of chronic herpes group virus infections in humans can be studied by longitudinally swabbing mucosal surfaces and sampling blood, helping to reveal the patterns of latency, reactivation, and dissemination, as well as giving insight into viral pathogenesis and host-pathogen interactions [16, 17]. Here, we present and analyse new oral EBV shedding data from a cohort of 85 Ugandan adults with and without HIV-1 coinfection. These data capture

EBV shedding dynamics in unprecedented detail, with saliva swabs being collected from participants daily and analysed for EBV presence and copy number. Although others have done such studies in developed countries, none exist with such a high degree of time resolution in Uganda, a country where EBV is often acquired much earlier in life than in developed countries [6–8]. Furthermore, while several previous studies have examined EBV mucosal shedding patterns in both HIV-1 uninfected and HIV-1 coinfected participants [5, 18–22], the majority have been in the setting of advanced HIV-1 infection or in participants receiving highly active antiretroviral therapy (HAART) [20–22]. Our data represent EBV shedding in HIV-1 coinfected individuals who are not receiving antiretroviral therapy and whose HIV-1 infection has not progressed to AIDS. Thus, our cohort participants maintain a relatively preserved immune system.

Using these data, we constructed and fit a new, stochastic mechanistic mathematical model describing infection dynamics within the crypts of the tonsillar epithelium. While some mathematical models of the within-host dynamics of EBV infection exist [10, 23–27], ours is the first implemented stochastically and the first fit to longitudinal data of EBV shedding. To fit these time series data, we present new methods based on the principles of Approximate Bayesian Computation. Furthermore, we uniquely use our model to examine how HIV-1 coinfection may influence EBV shedding patterns in the saliva. Previously, it has been hypothesised that increased EBV shedding with HIV-1 coinfection may be due to more frequent activation of EBV-infected B cells, leading to increased viral seeding of oral tissue and/or impaired T cell-mediated immune control of EBV replication, prolonging or inhibiting the clearance of infected epithelial cells [28–30]. To observe how these two mechanisms may influence EBV shedding, our model is fit to participant data, allowing the parameters controlling these mechanisms to vary. Thus, we estimated how both B cell reactivation and immune cell control contribute to EBV shedding patterns and examined how these parameters differ between individuals with and without HIV-1 coinfection. To verify these results, we repeated fitting using data collected from a similar cohort of 26 individuals from Seattle, Washington. Finally, we correlated model results to measures of HIV-1 loads, B cell activation factors, and CD4 T cell counts to further understand the relationship between model parameters and the impact of HIV-1 coinfection on EBV shedding.

## Results

### HIV-1 infection is associated with increased frequency and quantity of oral EBV shedding

Among our Ugandan cohort of 85 participants, 43 (51%) participants were HIV-1 seropositive. 32 participants (38%) were female and 53 (62%) were male, with a median age of 32 years (range 18–60 years). We collected a total of 2264 daily oral swabs, with a median of 29 swabs per participant (range 1–32). We show data on the EBV loads within these swabs for the first time here; however, additional details of the cohort have been previously published [31]. Other new data from this cohort, including EBV loads in genital swabs and plasma samples, can be found in S1 Text. Longitudinal EBV shedding in the saliva of all participants is shown in Fig 1. These data reveal the highly stochastic nature of EBV shedding in the saliva. While some participants transition between periods of continuous viral shedding to periods of no detectable shedding, others sustain detectable shedding throughout the entire time of observation. Furthermore, while HIV-1 coinfected participants generally have higher EBV viral loads than HIV-1 uninfected participants, some HIV-1 coinfected participants have uncharacteristically low viral loads and vice-versa, making it difficult to predict a participant's HIV-1 infection status from this data alone.

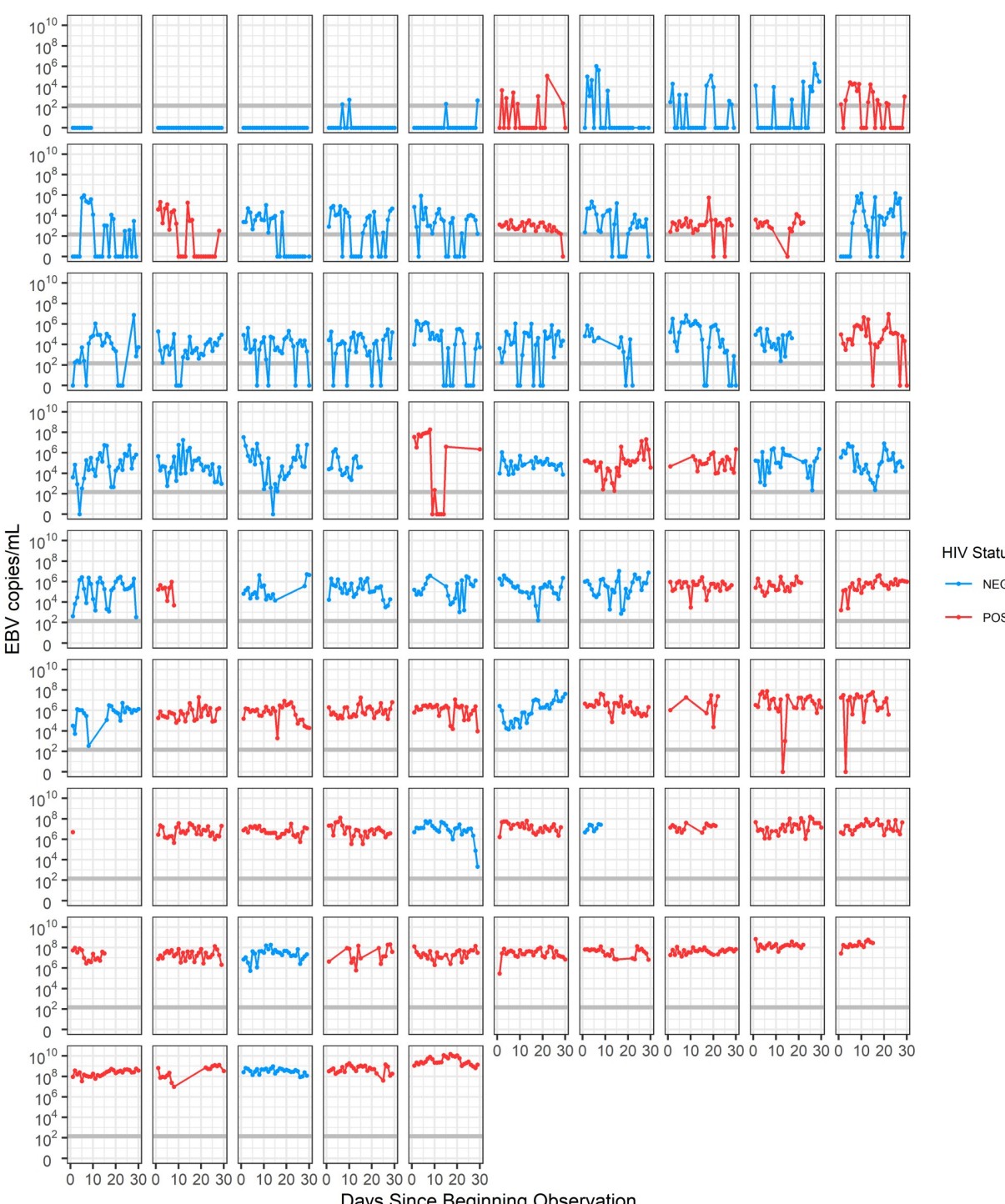

**Fig 1. Viral shedding patterns of study participants.** Viral loads in the saliva of study participants were measured daily via qPCR for a median of 29 days. Swabs up to 30 days post initiating observation are shown. Each plot represents the shedding pattern of a separate participant. Patients are arranged according to their median viral load. The grey horizontal line represents the threshold of EBV detection (150 EBV copies/ml).

We examined whether HIV-1 infection status affected the frequency of EBV detection in the saliva. We found that HIV-1 infection was significantly associated with EBV detection in oral swabs, increasing the frequency of observation 1.27-fold (CI = 1.10–1.47, p-value = 0.001, Fig 2A) and increasing the median $\log_{10}$ genome copies of EBV detected in oral swabs by 1.61 (CI = 1.29–1.93, p-value <0.001, Fig 2B). We also saw large variability in participants' viral loads over time, sometimes varying over 4 orders of magnitude in an individual participant (Fig 2C).

In addition to collecting data on EBV viral loads from participants, 72 participants had B cell activation factor (BAFF) levels in their serum measured, and all HIV-1 coinfected individuals had blood CD4+ T cell counts and HIV-1 RNA levels measured (Fig 2D). Mean BAFF levels were significantly higher in HIV-1 coinfected individuals than HIV-1 uninfected individuals (T-test, p = 0.01). Following linear regression, CD4+ T cell counts showed moderate negative correlation, and HIV-1 RNA showed strong positive correlation with median EBV viral loads in the saliva of HIV-1 coinfected individuals. BAFF levels showed moderate positive correlation with EBV viral loads in the saliva for HIV-1 uninfected individuals, and no correlation for HIV-1 coinfected individuals. Equations and correlation coefficients for this regression analysis are shown in Fig 2D.

## Mathematical model of EBV shedding in the tonsils

To obtain mechanistic insights into oral EBV shedding and to better understand the drivers of higher replication in HIV-1 coinfected individuals, we constructed a novel mathematical model that captures the relevant anatomic, virologic, and immunologic features of oral EBV infection. We built this model based on the structure of the oral tonsillar tissue where EBV is shed, known as Waldeyer's ring. In chronically infected individuals, EBV is shed in all areas of Waldeyer's ring, including the palatine, lingual, tubal tonsils, and adenoids [9]. Most of the tonsillar area is composed of stratified squamous epithelium or ciliated pseudostratified columnar epithelium, arranged into a series of crypts or folds, allowing for a large surface area [32]. The epithelium is often only one cell thick, allowing EBV to easily transcytose to reach the underlying lymphoid tissue where B cells and germinal centres are found [33]. The palatine tonsils have an estimated surface area of 295 cm$^2$, arranged into approximately 20 crypts [32], while the lingual tonsil area is composed of 35–100 crypts, and the adenoids are composed of a series of folds in lymphoid tissue [34]. By estimating that each palatine tonsil is approximately $\frac{1}{12}$ of the entire surface area of Waldeyer's ring, a series of 240 crypts, each serving as sites where EBV infection may occur, can represent the entire region. We assumed that the dynamics of each crypt are independent of each other and explicitly modelled the dynamics of infected epithelial cells, the immune response, and viral load within each crypt. In exploring this assumption of spatial independence, we found that having many spatially independent crypts was essential for reproducing the stochastic and highly variable nature of the participants' viral loads. In simulations without this spatial separation, viral loads and levels of immune surveillance within an individual's tonsils often equilibrated over time and no longer matched the stochastic nature of the data.

The dynamics within each crypt are shown in Fig 3. We assumed that latently infected B cells that are circulating throughout the body return to Waldeyer's ring, reactivate, and infect the tonsillar epithelium at a constant rate $b$. Infected epithelial cells, $I$, infect other epithelial cells through cell-to-cell contact at a constant per-capita rate $\beta$, making the simplifying assumption that target cell number is not limiting. This assumption can be justified because of the large number of epithelial cells that are expected to make up Waldeyer's ring. Assuming the tissue composition across the tonsils is similar to that in the palatine tonsils where the most

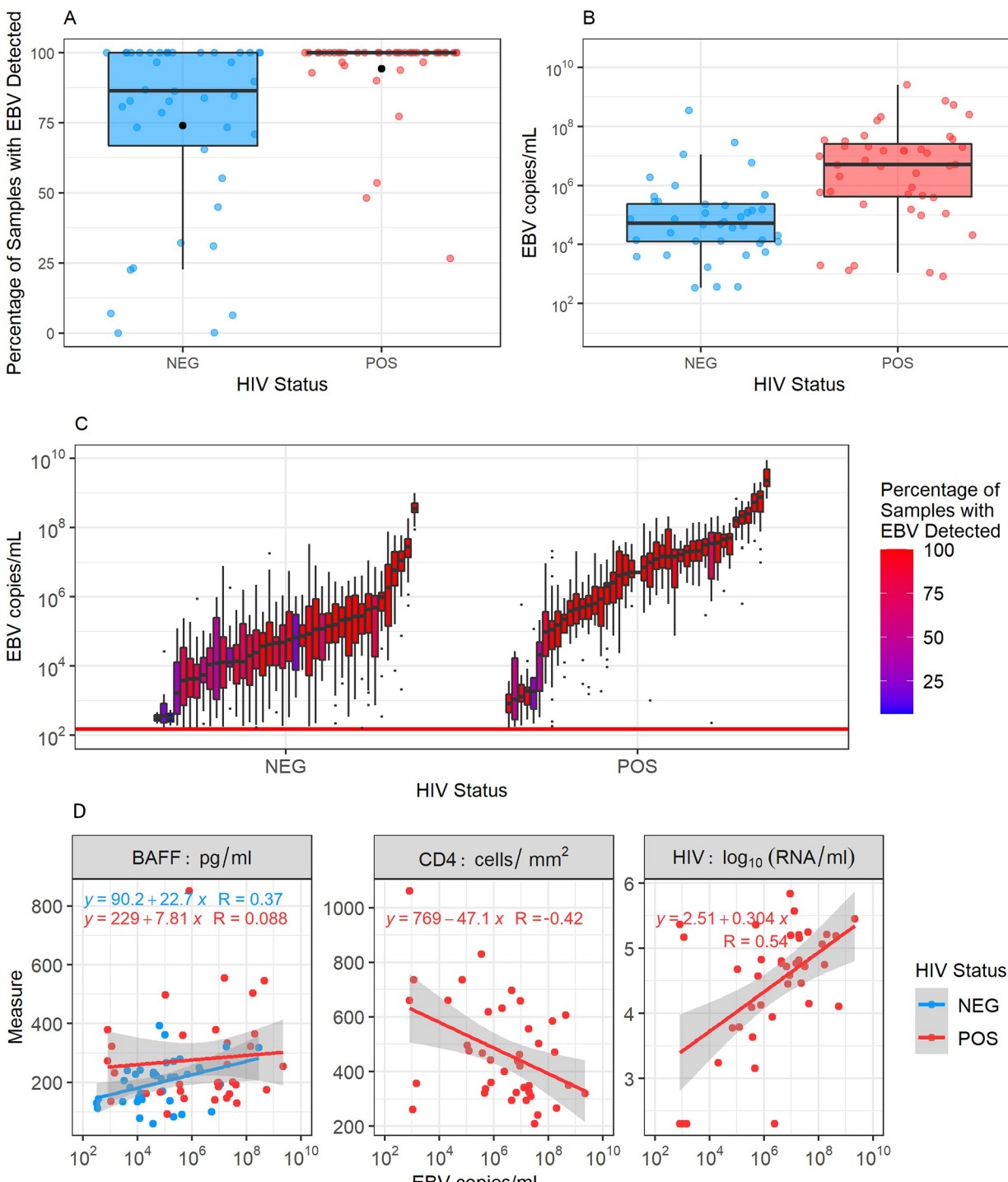

**Fig 2. Impact of HIV-1 coinfection on oral EBV replication.** A. Percentages of saliva samples that tested positive for EBV for each participant. Black dots indicate the percentage of samples that tested positive for EBV when pooling participant samples. In HIV-1 uninfected participants, the median percentage of swabs positive for EBV was 86% (range 0–100%, interquartile range (IQR) 33%), while in HIV-1 coinfected participants, the median percentage of swabs positive for EBV was 100% (range 27–100%, IQR 0%). B. Median EBV viral loads/ml in oral swabs testing positive for EBV for each participant. All graphs stratify participants by HIV-1 infection status. Coloured dots show the median value of the statistic for each participant, while bars and whiskers show the spread across participants. C. Distributions of participants' oral swab viral loads. Each box and whisker represents the viral loads of EBV-positive oral swabs for an individual participant. The percentage of oral swabs that tested positive for EBV for each participant is indicated by the colour of the box. Of swabs that tested positive for EBV, viral loads varied over time by a median of 3.49 orders of magnitude (range 0.34–5.27, IQR 1.32) within individual HIV-1 uninfected participants and a median of 2.30 orders of magnitude (range 0.95–5.93, IQR 1.15) within individual HIV-1 coinfected participants. The red horizontal line represents the threshold of EBV detection (150 EBV copies/ml). D. Participants' BAFF levels, CD4+ T cell counts, and HIV-1 RNA loads correlated against median EBV copies/ml of saliva.

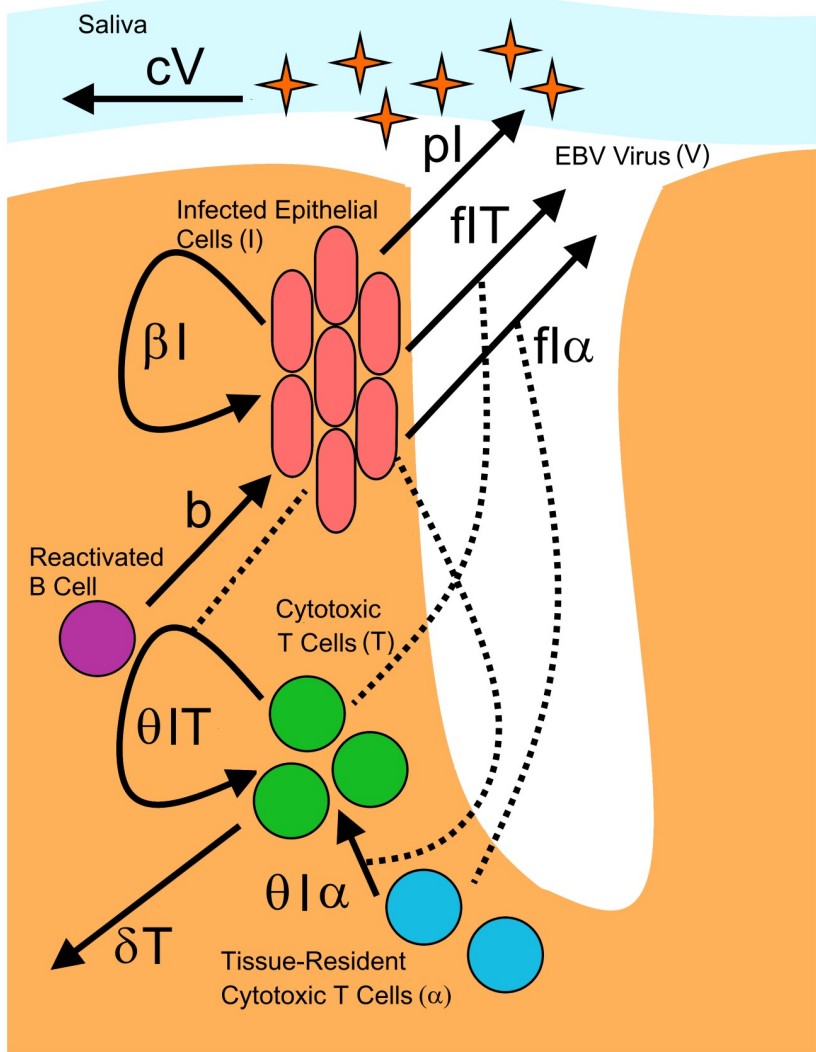

**Fig 3. Description of single crypt dynamics.** Waldeyer's ring is represented as a series of 240 individual crypts in which infection dynamics occur. Within each crypt, the population dynamics of infected epithelial cells (I), cytotoxic T cells (T) and EBV (V) are described. The viral load detected in saliva is represented by the total virus aggregated across all crypts.

detailed information about Waldeyer's ring is known, the total surface area of Waldeyer's ring would be 3540 cm$^2$. While epithelial cell densities within tonsils have not been measured, a previous estimate of epithelial cell densities in the genital tract where herpes simplex lesions can form is $1.7 \times 10^6$/cm$^2$ [35]. Assuming similar cell densities in the tonsils, this would indicate that a total of $6.1 \times 10^9$ epithelial cells are present in Waldeyer's ring. With the production rate of virions per day per cell likely around $10^4$ [10, 24], and maximum viral loads seen in cohort participants being just under $10^{10}$ (Fig 1), this would indicate that at most, $10^6$ infected cells would be needed to produce the viral loads seen in participants. As our estimates for the total number of epithelial cells within Waldeyer's ring are multiple orders of magnitude larger than this, it is unlikely target cell number would ever be a limiting factor.

Epithelial infection causes the recruitment and proliferation of EBV-specific cytotoxic T cells, $T$, at a per-capita rate $\theta I$. Cytotoxic T cells kill infected epithelial cells following the law of

mass action at a rate $fIT$. We assumed that cytotoxic T cells die or leave the tonsils at a per-capita rate $\delta$. Independent of infection, we assumed a constant number of EBV-specific cytotoxic T cells, $\alpha$, are tissue-resident. Like $T$, these cells can kill infected epithelial cells and stimulate the proliferation of new EBV-specific cytotoxic T cells; however, while population $T$ leaves the system over time, these tissue-resident T cells remain within the tissue and do not recirculate [36, 37]. Including these tissue-resident T cells in the model means there are always immune cells present to respond to new infection, and tissue is never entirely unprotected, allowing for faster control of infection. EBV virions, $V$, are produced by infected epithelial cells, enter saliva at a per-capita rate $p$ and are cleared at a per-capita rate $c$. In this model, we assumed the main contributors to virus in the saliva are infected epithelial cells. Thus, we did not directly model virions produced by infected B cells [10]. With the propagation of EBV infection shown to be 800-fold more efficient through cell-to-cell contact rather than through free virus, we also chose to assume all new epithelial cell infection is caused by cell-to-cell contact [10, 38].

The concentration of EBV detected in the saliva of participants was highly variable, and frequently undetectable. Therefore, we chose to implement our model in a stochastic framework in order to capture these traits (Methods). Our model assumptions were used to build a chemical master equation system [39] that describes all system reactions within a single crypt, as follows:

$$I \rightarrow I + 1 \qquad \text{with rate } b + \beta I \qquad (1)$$

$$I \rightarrow I - 1 \qquad \text{with rate } fI(T + \alpha) \qquad (2)$$

$$T \rightarrow T + 1 \qquad \text{with rate } \theta I(T + \alpha) \qquad (3)$$

$$T \rightarrow T - 1 \qquad \text{with rate } \delta T \qquad (4)$$

$$V \rightarrow V + 1 \qquad \text{with rate } pI \qquad (5)$$

$$V \rightarrow V - 1 \qquad \text{with rate } cV. \qquad (6)$$

By implementing this model stochastically, we also allow for infection to potentially die out in a crypt. With the seeding of infection by EBV-infected activated B cells, CD8 T cells will quickly proliferate and begin to control infection within the developing plaque. If enough CD8 T cells are present, infection may be cleared within this crypt. This crypt may then remain clear of infection until EBV-infected B cells are stochastically chosen to reactivate and seed new infection.

When considering the parameters of this model, many could potentially show small variations between individuals; however, most would likely not change based on HIV-1 infection status. Thus, to simplify the fitting of our model, we fixed many of these parameters to set values based on those found in the literature. While more cells may become infected with EBV if an individual has HIV-1 coinfection, once a cell becomes infected, it is unlikely that the rate at which that cell produces EBV ($p$) would change based on HIV-1 infection status. Similarly, EBV that is released by an infected cell and enters saliva likely maintains a constant clearance rate ($c$). While HIV-1 infection is known to impact innate immunity in the oral mucosa, which could impact how long a virus survives outside the cell, the overall phenomenon is poorly understood, and we assume this impact is small [40]. The natural death rate of EBV-cytotoxic T cells ($\delta$), the ability of each of these cells to target and kill an infected cell ($f$), and the number of these cells that remain tissue-resident are again likely not impacted by HIV-1 infection

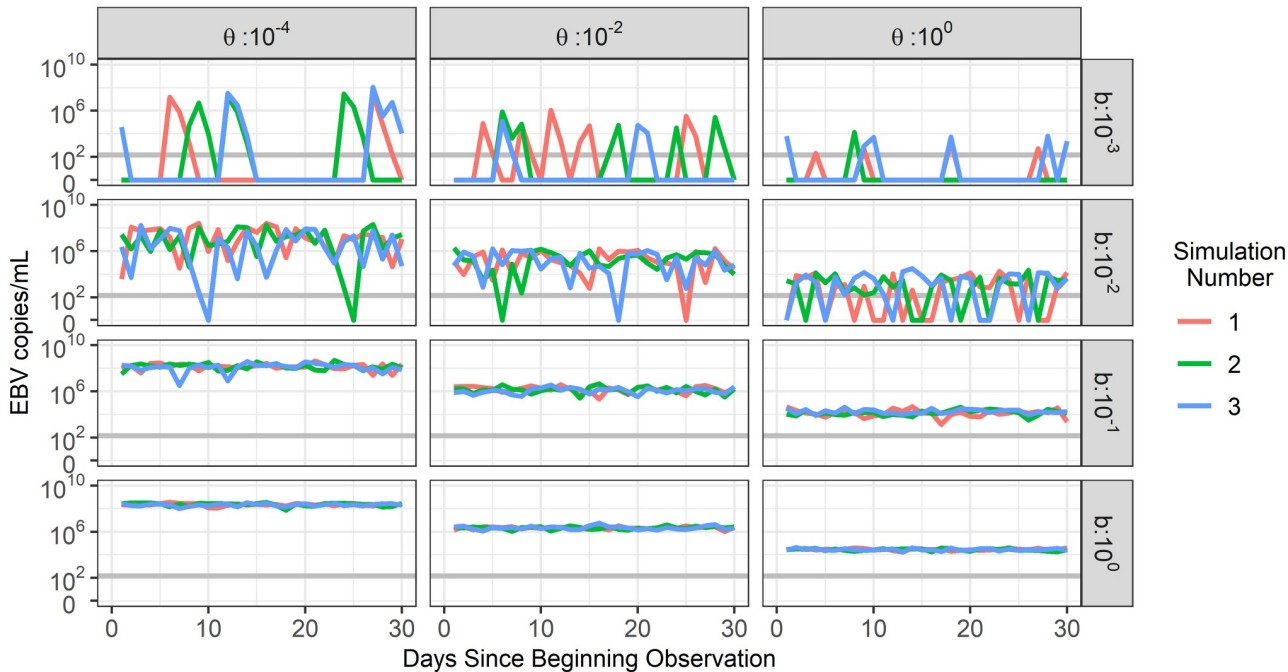

**Fig 4. The impact of varying $b$ and $\theta$ on model simulation trajectories.** Model simulations for different pairings of values for parameters $b$ and $\theta$ are shown (units of cell day$^{-1}$ and day$^{-1}$ respectively). Simulations reproduce the stochastic nature of the data and are able to capture a wide variety of EBV shedding traits. For all simulations, $\beta = 50$ day$^{-1}$, $f = 0.1$ day$^{-1}$cell$^{-1}$, $\alpha = 200$ cells, $\delta = 0.1$ day$^{-1}$, $p = 10^4$ virions day$^{-1}$ ml$^{-1}$ cell$^{-1}$, and $c = 6$ day$^{-1}$. The grey horizontal line represents the qPCR threshold of detection. All simulated viral loads below this threshold were set to zero to match with participant data.

status. Thus, these parameters remained fixed throughout our fitting and analysis. However, based on previous studies [28–30], we hypothesized that coinfection is likely to influence the rate at which infected B cells reactivate since HIV-1 antigen could stimulate this reactivation. Similarly, since HIV-1 coinfection often leads to a dampened ability to control other infections, HIV-1 coinfection could influence the rate at which EBV-specific cytotoxic cells are recruited and proliferate at the site of infection. Thus, we assumed one or both of these processes were responsible for the differences seen in EBV shedding patterns in individuals of different HIV-1 coinfection statuses. To determine which of these two processes is most important to the phenomenon, or whether both contribute, parameters $b$ or parameter $\theta$ were left free to be fit to participants data, while other parameters remained fixed.

A sensitivity analysis for all parameters was initially performed (Fig A in S2 Text). As a result, fixed parameter values in the model were set to $\beta = 50$ day$^{-1}$, $f = 0.1$ day$^{-1}$cell$^{-1}$, $\alpha = 200$ cells, $\delta = 0.1$ day$^{-1}$, $p = 10^4$ virions day$^{-1}$ ml$^{-1}$ cell$^{-1}$, and $c = 6$ day$^{-1}$. Examples of model simulation trajectories with these fixed values and different values of $b$ and $\theta$ are shown in Fig 4. Low viral loads in the saliva are achieved with a high value of $\theta$ and a low value of $b$, while high, sustained viral loads are acheived with a low value of $\theta$ and a high value of $b$. High values of $b$ also allow for a more constant level of virus to be detected in saliva while lower values of $b$ create more variance in viral load due to less frequent reactivation of latently-infected B cells and seeding of new infection within the tonsils.

To determine what pairings of $b$ and $\theta$ produce simulations that best reproduce each participant's data, parameters $b$ and $\theta$ were fit to each participant's data through Approximate Bayesian Computation. Briefly, we first selected a participant and calculated summary

statistics capturing the nature of their EBV shedding patterns. Varying parameters $b$ and $\theta$ and running model simulations, we identified what values of $b$ and $\theta$ best reproduced the summary statistics of the participant's data. The parameter values lending to good fits were used to build a posterior distribution for parameters $b$ and $\theta$ for the selected participant. This process was repeated for each participant, leading to different distributions of parameters $b$ and $\theta$ for each participant. These distributions were then combined using importance sampling to infer how parameters of different groups of individuals varied. Further details on these analyses can be found in the Methods and S2 Text. Examples showing how well simulation runs fit to participant data can be found in Fig B in S2 Text. Among all 85 participants' data, we were able to fit parameters to 82. Of the 3 participants whose data could not be fit, 2 participants had no EBV detected in any of their saliva swabs, and 1 participant had only 1 swab collected.

## Greater oral EBV shedding with HIV-1 coinfection is due to both increased B cell reactivation and weaker cellular immune response

Once all participant data was fit to our model, we examined how the cumulative distributions of parameters $b$ (rate of B cell reactivation causing new lytic epithelial infection) and $\theta$ (rate of EBV-specific cytotoxic T cell proliferation and recruitment) differed between individuals. Parameter distributions stratified by HIV-1 infection status and median EBV viral load in saliva are shown in Fig 5.

A randomly selected HIV-1 coinfected individual is expected to have a higher $b$ than a randomly selected HIV-1 uninfected individual with probability 0.76, and overall the median value of $b$ is 2.9 times higher in HIV-1 coinfected individuals than in HIV-1 uninfected individuals. Similarly, an HIV-1 coinfected individual has a lower $\theta$ than an HIV-1 uninfected individual with probability 0.74, and the median value of $\theta$ is 19.7 times lower in HIV-1 coinfected individuals.

While these results indicate both $b$ and $\theta$ differ based on HIV-1 infection status, there is substantial overlap in these distributions due to the wide variety of shedding patterns observed across participants and the sometimes similar shedding patterns seen between HIV-1 uninfected and coinfected participants. Thus, we stratified $b$ and $\theta$ by participants' median EBV viral load in saliva to better reveal a pattern. When stratified in this way, differences in the distributions of $b$ and $\theta$ for each group become clearer, with individuals belonging to a specific median viral load group having distinct $b$ and $\theta$ values. This result indicates that the values of parameters $b$ and $\theta$ may be better explained by an individual's EBV viral load rather than their HIV-1 infection status. Details on these distributions are shown in Fig 5C and 5D.

Because both $b$ and $\theta$ appear to explain differences between groups of individuals, we sought to quantify the correlation between these two parameters (Fig 6). When looking at the within-group correlation of participants with similar median viral loads, the $b$ and $\theta$ values selected during fitting have a moderate positive correlation, indicating that $b$ and $\theta$ can counter-balance to produce similar viral loads. However, when observing all data, fit $b$ and $\theta$ values are moderately negatively correlated, indicating the individuals with the highest viral loads are those whose parameters feature high $b$ and low $\theta$ values. It can also be noted that some individuals with median viral loads of $10^2 - 10^4$ have higher $b$ values than is strictly predicted from these trends. This cluster on the graph comes from individuals who are HIV-1 coinfected with low viral loads (thus a high $\theta$) but a high frequency of positive swabs (higher $b$).

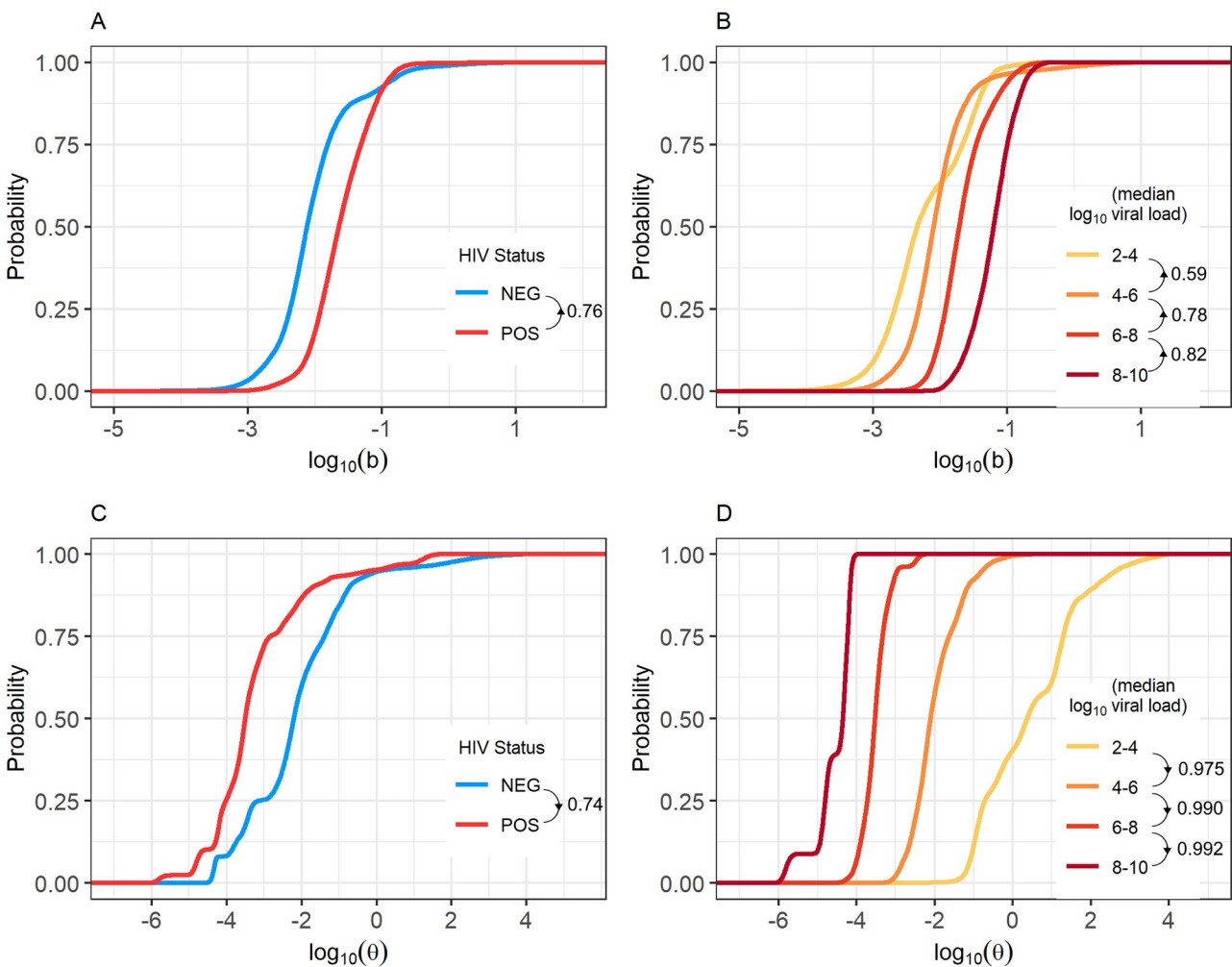

**Fig 5. Cumulative distribution of parameters *b* and *θ*, stratified by HIV-1 status and median EBV viral load in saliva.** Fitting our mathematical model to participant data revealed that parameter *b* is usually greater in HIV-1 coinfected participants (A) and increases with median EBV viral load (B). Parameter *θ* is usually lower in HIV-1 coinfected participants (C) and decreases with median saliva EBV viral load (D). Directional arrows and numbers by figure legends indicate the probability that a randomly selected individual of one group has a higher parameter value (be it *b* or *θ*) than a randomly selected individual in a second group. Arrows show the direction of comparison.

## EBV infection within tonsillar crypts behaves differently in HIV-1 coinfected and HIV-1 uninfected individuals

Using the results of our model, we next examined how simulations predicted the distribution of infected cells throughout the different tonsillar crypts of our cohort participants (Fig 7).

The cumulative distributions for the median number of crypts with an active infection at any given time for HIV-1 coinfected and HIV-1 uninfected individuals are shown in Fig 7A. Despite all crypts within an individual being governed by the same set of parameters, crypt dynamics are not uniform, with usually only a few crypts producing virus at any given time. Over time, an HIV-1 coinfected person will have a higher median number of actively infected crypts than an HIV-1 uninfected person with probability 0.68 (Fig 7A). While these distributions have large variance (IQR of 1 and 4 for HIV-1 uninfected and HIV-1 coinfected individuals, respectively), over time HIV-1 uninfected participants are expected to have a median of 1 crypt within their tonsils actively producing virus, while HIV-1 coinfected individuals are

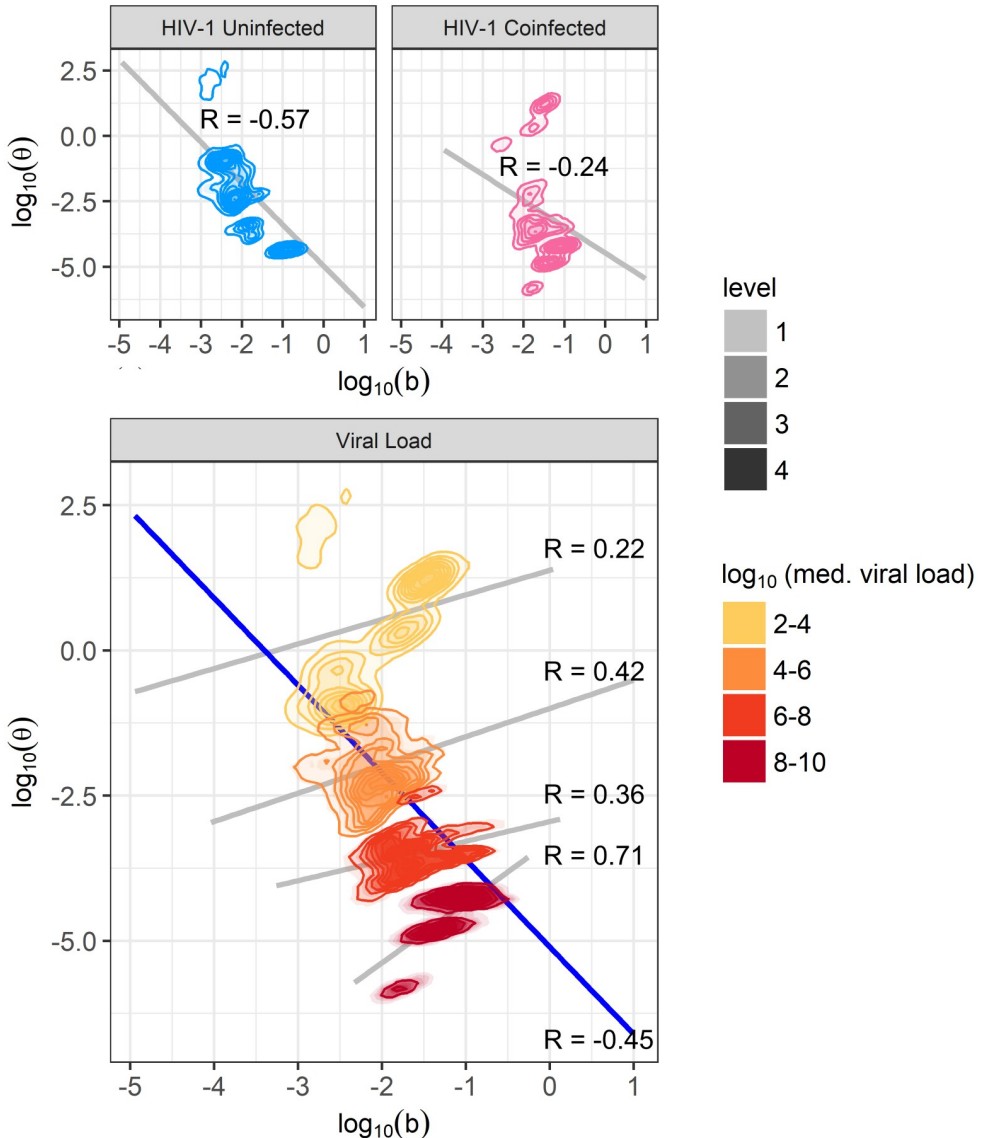

**Fig 6. Correlation between parameters *b* and *θ*.** Obtained densities of parameters *b* and *θ* are plotted, stratified by HIV-1 infection status and median oral EBV viral load group. Across all participants, or when stratifying by HIV-1 infection status, *b* and *θ* are negatively correlated (grey lines in top plots and blue line in bottom plot). However, since *b* and *θ* have opposite effects on viral load, positive correlations are seen within each viral load group (grey lines, lower plot).

expected to have 2. Thus, our results indicate that rather than all crypts always actively infected and generating low amounts of virus, the viral loads seen in saliva are more often generated by only a few actively infected crypts, each producing higher amounts of virus. These results match well with previous estimates that indicate individuals have ≤ 3 independent plaques of oral epithelial infection at any given time [10].

Our results also suggest that infected crypts within an HIV-1 coinfected individual produce more virus than the crypts of an HIV-1 uninfected individual. Over time, an HIV-1 coinfected individual is expected to have a higher median viral load within their actively infected crypts than an HIV-1 uninfected individual with probability 0.65 (Fig 7C). Our simulations predict a

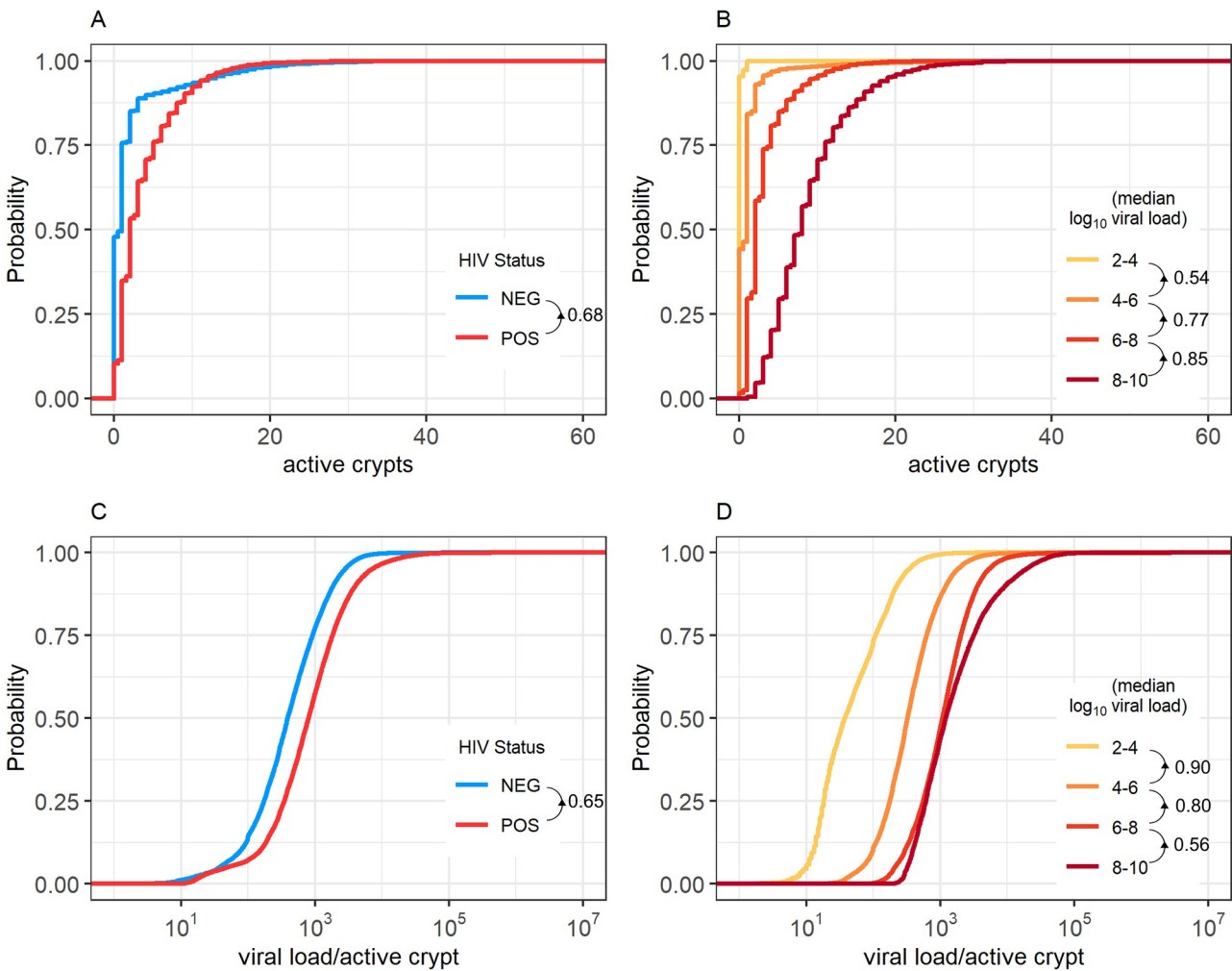

**Fig 7. Predicted numbers of active crypts and viral load per active crypt.** Cumulative distributions of the median number of crypts actively producing EBV (A and B) and the median EBV viral load produced by an active crypt at any given time (C and D) are shown stratified by participant HIV-1 status and EBV median viral load in the saliva. Increases in median salivary EBV viral load are caused by a higher number of crypts having active (B) and more extensive (D) infection. This trend translates to HIV-1 coinfected participants having more infected crypts infected, and each infected crypt producing more virus. We see that HIV-1 uninfected individuals usually have more actively infected crypts and more virus per active crypt than HIV-1 coinfected individuals. Similarly, we see that individuals with higher median viral loads in their saliva usually have more actively infected crypts and more virus per active crypt than individuals with lower median viral loads. Directional arrows and numbers by figure legends indicate the probability that a randomly selected individual of one group has a higher parameter value (be it the number of active crypts or the viral load per active crypt) compared with a randomly selected individual in a second group. Arrows show the direction of comparison.

median of 798 EBV DNA copies per active crypt in HIV-1 coinfected individuals and 389 EBV DNA copies per active crypt in HIV-1 uninfected individuals. These distributions again have large variance, leading to overlap (IQR of 753 and 1472 for HIV-1 uninfected and HIV-1 coinfected individuals, respectively).

We also examined the behaviour of traits when stratifying our simulations according to participants' median viral loads in the saliva. Higher median EBV viral loads detected in saliva correlate well with higher numbers of actively infected crypts (Fig 7B) and higher viral loads per actively infected crypt (Fig 7D). Together, these results indicate that high EBV loads in the saliva are caused by more frequent and extensive infection in tonsillar crypts.

## Validating model results using participant biological data

Because our model predicted that increased EBV loads are caused by both increased B cell reactivation and decreased immune control within tonsillar crypts, we wanted to evaluate whether these predictions could be directly validated by laboratory data. Thus, we assessed whether the CD4+ T cell count and HIV-1 RNA copy number in HIV-1 coinfected participants and the level of BAFF present in the blood of all participants matched our model's predictions. While none of these values directly correspond to parameters within our mathematical model, CD4+ T cell count should relate to parameter $\theta$, while BAFF levels should relate to parameter $b$. When looking at the relationship between these values and parameters $\theta$ and $b$, in all scenarios these values correlated in the expected direction, further supporting results (Table 1). A generalized linear model (Methods) revealed that each $\log_{10}$ increase in HIV-1 RNA copies/ml significantly decreased the predicted value of $\theta$ but did not significantly affect $b$. Each 100-cell/mm$^3$ increase in CD4+ count significantly changed the value of $\theta$ and $b$, increasing $\theta$ and decreasing $b$. Lastly, each 100 pg/ml increase in BAFF caused a nearly-significant change in the value of $b$, increasing its value, while causing a non-significant decrease in $\theta$.

We repeated this analysis to observe whether CD4+ T cell counts, HIV-1 plasma RNA levels, and BAFF levels correlated with the median amount of virus detected in positive swabs (Table 1). CD4+ T cell count and BAFF levels significantly correlated with the amount of EBV detection in oral swabs. Each additional 100-cell/mm$^3$ increase in CD4+ count was associated with a 68% reduction in the amount of virus detected, consistent with cell-mediated immunity conferring partial but incomplete control of EBV replication. Each 100 pg/ml increase in BAFF was associated with a 71% increase in the amount of virus detected, supporting our hypothesis that higher B cell activation increases EBV shedding. HIV-1 viral loads significantly correlated with the amount of virus detected in samples, with each $\log_{10}$ increase in HIV-1 RNA correlating with a 569% increase in the amount of EBV detected in saliva swabs.

## Application of our model to an independent data set from a North American cohort

To test its generalizability, we applied our model to a previously described set of data from a cohort of 26 participants in Seattle, Washington, who underwent daily oral EBV sampling and

**Table 1. Effects of plasma HIV-1 load, CD4+ T cell count and BAFF amount on the Ugandan cohort participants' median viral load and median values of parameters $b$ and $\theta$.** In cohort participants, median viral load detected via qPCR and values of parameters $b$ and $\theta$ are influenced by the CD4+ T cell count, HIV-1 plasma viral load, and the amount of BAFF in serum. The fold-change (FC) in participants' median viral load, or model-fit $b$ and $\theta$ values for every $\log_{10}$ increase in HIV-1 RNA copies/mL, every 100 CD4+ T cell/mm$^3$ increase, and every 100 pg/ml increase in BAFF is shown. Note that data on CD4+ T cell count and HIV-1 RNA was only available for HIV-1 coinfected participants.

| Trait 1 | Trait 2 | FC | 95% CI | p-value |
|---------|---------|-----|--------|---------|
| HIV-1 RNA | viral load | 6.69 | 2.53–17.71 | <0.001 |
|  | $\theta$ | 0.16 | 0.06–0.42 | <0.001 |
|  | $b$ | 1.21 | 0.89–1.63 | 0.226 |
| CD4+ T cell | viral load | 0.42 | 0.24–0.73 | 0.004 |
|  | $\theta$ | 1.84 | 1.05–3.22 | 0.040 |
|  | $b$ | 0.80 | 0.69–0.92 | 0.004 |
| BAFF | viral load | 1.71 | 0.92–3.20 | 0.096 |
|  | $\theta$ | 0.62 | 0.33–1.19 | 0.156 |
|  | $b$ | 1.22 | 0.99–1.49 | 0.061 |

testing using the same methods as the Ugandan cohort described above [19]. A total of 1323 swabs were collected during the 8-week period of the study, with a median of 55 swabs per participant (3–61 swabs). Of these participants, 16 (62%) were HIV-1 coinfected and, if on HAART, were required to remain on a stable regimen throughout the study. None were receiving other antiviral drugs at the time of enrollment. While the objective of the Seattle study was to analyze the effects of valganciclovir on daily EBV oral shedding, we restricted our analysis to the data from the eight-week period when participants received a placebo.

Participants of the Seattle cohort showed significantly lower oral EBV viral loads than those of the Uganda cohort. Among Seattle cohort participants, HIV-1 uninfected participants had a mean of 1.4 $\log_{10}$-lower EBV viral loads in their positive swabs than participants who were HIV-1 uninfected in the Uganda cohort (p-value<0.001). HIV-1 coinfected participants in the Seattle cohort had a mean of 2.0 $\log_{10}$-lower viral loads in their positive swabs than those who were HIV-1 coinfected in the Uganda cohort (p-value<0.001). Nonetheless, both studies provide robust assessments of EBV shedding, and we expect the host-pathogen interactions occurring within the tonsils to be the same. Our mathematical model fit the Seattle cohort data again with high fidelity (Fig B in S3 Text) and produced similar results in terms of the number of infected crypts, virus produced by crypts, and the values for parameters $b$ and $\theta$ when stratified by HIV-1 status and median EBV load (Figs C and D in S3 Text). We give full details in S3 Text. Thus, the Seattle cohort data supports our modelling approach and findings from the Uganda cohort.

## Discussion

By capturing the replication patterns of EBV in the saliva of HIV-1 coinfected and uninfected Ugandan cohort participants, we were able to develop a novel stochastic mathematical model of EBV infection in the tonsils and use it to evaluate potential explanations for why individuals with HIV-1 coinfection have higher EBV viral loads and are more susceptible to EBV-related malignancies. Specifically, our model indicates that increased oral EBV shedding with HIV-1 coinfection is due to both greater reactivation of EBV-infected B cells as well as impaired EBV-specific cytotoxic T cell immune control.

Previous literature has reported that individuals infected with HIV-1 have higher oral EBV shedding than HIV-1 uninfected individuals [3, 4, 20–22, 41, 42]. However, few data sets present as detailed a representation of the dynamics of oral EBV shedding as we have shown here or show how EBV shedding behaves in adults not receiving any antiviral or antiretroviral treatment. From our analysis of the data, we found that HIV-1 infection is associated with a significant increase in both the frequency and quantity of oral EBV shedding in Ugandan adults with chronic EBV infection. Further, there was a statistically significant association between higher CD4+ T cell counts in the blood of HIV-1 coinfected participants and a lower frequency of EBV shedding in their saliva.

While previous mathematical models have examined the within-host dynamics of EBV infection [10, 23–26], none have examined the differences between HIV-1 coinfected and HIV-1 uninfected individuals [35, 43–45]. Strengths of our approach include the incorporation of granular quantitative EBV shedding measurements from two independent cohorts, each made up of HIV-1 coinfected and uninfected individuals, and the inclusion of CD4+ T cell counts and HIV-1 plasma viral load data from coinfected participants, as well as measurements of the B cell activation marker BAFF in the blood of all participants. Limitations include the lack of data on EBV-specific T cell responses.

Our novel mathematical model is based on representing the tonsillar epithelium as a series of crypts, each serving as a potential site of epithelial infection and viral shedding. In this way,

a single tonsillar crypt behaves similarly to how individual herpes simplex virus (HSV)-2 lesions have been modelled in the past, with reactivation of EBV-infected B cells being analogous to the release of HSV-2 from infected neurons, sparking new epithelial lesions [35, 43–45]. These previous models all captured the stochastic patterns of HSV-2 shedding well, which appear similar to the patterns of oral EBV shedding. Our model assumes that all tonsillar crypts are independent of one another. In reality, virus from one infected crypt may spill over and seed infection in a neighbouring crypt, rather than EBV entering an uninfected crypt purely via the reactivation of B cells as we assume in our model. By not accounting for this, our predicted B cell reactivation rates are likely higher than their true biological values. However, assuming the amount of viral spill-over into neighbouring crypts is proportional to the viral load, our qualitative comparison remains valid. Furthermore, modelling the tonsillar crypts as multiple, segregated sites of infection was essential for simulating the high variability in viral load seen over time in cohort data. When the tissue was treated as one well-mixed region, viral loads and immune cell counts would equilibrate and lose the high variability seen in participant data. This result indicates that while virus and immune cells may travel between crypts, this effect is likely minimal. While a fully spatial model including the mobility of virus and immune cells throughout the tonsils would have been ideal, we could only have parameterized it speculatively, and it would be computationally intensive. Our strategy of a crypt-level simulation is simple enough to be computationally feasible but complex enough to retain the inherent stochasticity and spatial diversity of EBV infection dynamics within the tonsils.

Our mathematical model was fit explicitly, assuming that differences in EBV shedding patterns were either due to changes in the rate of B cell activation, the rate of immune cell proliferation and recruitment within tonsillar crypts, or both. While some may consider it preferable to have a model where all parameters are rigorously fit, the large number of parameters in our model prevents this when using a stochastic model. Further, most parameters would likely remain unaffected by HIV-1 coinfection. We specifically chose to fit the two parameters most likely to be influenced by HIV-1 infection status. Indeed, B cell activation and plasma cell differentiation have been shown to induce EBV reactivation [12] and increased B cell activation is associated with HIV-1 infection [30, 46]. Further, HIV-1 has long been known to hinder immune control of coinfections. By explicitly focusing on these two factors, we were able to determine if they alone could explain the differences seen between HIV-1 uninfected and HIV-1 coinfected EBV shedding patterns. The good fits between the model and data indicate that indeed they can. We found that both factors contribute to higher EBV viral loads in HIV-1 coinfected individuals, and were able to determine distributions for these parameters, which have previously never been calculated for EBV. However, it is difficult to discern from our model whether the effects of HIV-1 infection on B cell activation or the immune response are more important in determining EBV shedding dynamics as both showed similar changes based on HIV-1 infection status. As such, further experimental work or study of cohort participants is needed to address this question.

Our model also revealed interesting predictions on how infection is distributed amongst the crypts of the tonsils. Our model predicts that HIV-1 coinfected individuals have more actively infected crypts that each produce more EBV at any given time than HIV-1 uninfected individuals. These results are consistent with previous observations that cellular immune control of EBV infection in HIV-1 coinfected individuals is impaired [47].

While our mathematical model describes within-host EBV infection dynamics, it does not provide information on how HIV-1 coinfection may affect the transmission rate of EBV among a population. Theoretically, higher EBV loads caused by HIV-1 coinfection should lead to greater risks of transmission. Future work aimed at linking our model to an epidemiological one describing EBV spread within a population could help elucidate how dampened immune

responses or greater activation of latently infected B cells caused by HIV-1 coinfection affect transmission and further strengthen the importance of understanding these within-host dynamics.

Importantly, our results have implications for strategies to prevent EBV infection and disease. EBV-specific cellular immunity is recognized as critical for controlling EBV replication and preventing EBV-associated malignancies. [47–52]. Independent of restoring EBV-specific cellular immune responses, strategies to reduce B cell reactivation in EBV-infected participants might limit viral replication, transmission, and related malignancies [53, 54].

## Methods

### Ethics statement

All participants within the Seattle Cohort provided written informed consent for study participation. Participants within the Uganda cohort provided either written or verbal informed consent, with verbal consent being obtained in place of written consent when participants were unable to read or understand the consent form. Verbal consent was documented via a thumb-print from the participant on the oral consent form along with a signature from the person obtaining consent. Study procedures for both human participant cohorts, including written and verbal consent procedures, were approved by the University of Washington Human Subjects Review Board (UW IRB no. 27014 and UW IRB no. 02–1500-B 04 for the Uganda and Seattle cohort, respectively). Additional approval for the Ugandan cohort study was given by the Makerere University Research and Ethics Committee and the Uganda National Council for Science and Technology.

### Cohorts and samples

Men and women aged 18 to 65 were enrolled in the Uganda cohort as previously described and were followed for four weeks [31]. Eligible HIV-1 seropositive participants had a CD4+ T cell count greater than 200 cells/mm$^3$ and were not taking antiretroviral therapy, following the WHO guidelines at that time [55]. Men aged 24 to 66 were enrolled in the Seattle cohort as previously described [18, 19]. As the Seattle cohort shedding data was obtained from a randomized placebo-controlled cross-over trial of valganciclovir, only data collected while participants were receiving placebo were used for this study. Both participants and pill administrators were unaware of group assignments. Participants of both cohorts did not take any drugs with anti-herpesvirus activity during the study period. Self-collected daily oropharyngeal swabs for both cohorts were collected by swabbing the oral mucosa and pharynx with a Dacron swab and were then placed in a vial containing 1 ml of 1X digestion buffer, stored at room temperature, and returned at weekly (Ugandan cohort) or bi-weekly (Seattle cohort) clinical visits. In the Uganda cohort, focused physical exams and collection of genital and plasma samples were performed at weekly clinic visits. These data are described in the S1 Text. All data collected and used in this study is stored in the Dryad data repository: https://doi.org/10.5061/dryad.w6m905qkh [56].

### Laboratory testing

Commercially available immunoassays were used to ascertain HIV-1 and EBV serostatus (Inverness Medical Innovations, Inc and Wampole® for the Ugandan cohort and Abbott Laboratories for the Seattle Cohort [19]). For the Ugandan cohort, CD4+ T cell counts and plasma HIV-1 RNA levels were determined in HIV-1-infected participants at the Makerere University-John Hopkins University laboratory using standard cell sorting techniques and the

Amplicor HIV-1 monitor test (Roche, version 1.5), respectively. For both cohorts, DNA was extracted from mucosal swabs and plasma [57], and real-time quantitative polymerase chain reaction (qPCR) was performed using specific primers to detect EBV [58], with positive and negative controls as previously described [16, 57]. Mucosal samples with greater than 150 copies/ml and plasma samples with greater than 50 copies/ml herpesvirus DNA/ml were considered positive [59]. Following these tests, 72 participants had enough plasma remaining (0.5 ml) to have levels of soluble B cell activating factor (BAFF) measured using the Human BAFF DuoSet enzyme-linked immunosorbent assay (ELISA) kit (R&D systems). ELISAs were performed using the sandwich technique and done in duplicate with the averages used in the analysis.

## Statistical analyses of data

The frequency of mucosal shedding and viremia was defined as the proportion of samples testing positive for EBV. The frequency of mucosal shedding was first compared in HIV-1 coinfected and uninfected participants. To do this, we used generalized estimating equations (GEE) and assumed frequencies followed a Poisson distribution. Frequencies of mucosal shedding and viremia were also modelled with GEE allowing for continuous adjustment for each 100 cell/mm$^3$ increase in CD4+ T cell count, each $\log_{10}$ increase in HIV-1 RNA, and each 100 pg/ml increase in BAFF. Again, we assumed that the frequency of shedding follows a Poisson distribution. For these models, BAFF measurements were available for all participants, while CD4+ T cell counts and HIV-1 RNA loads were only available for HIV-1 infected participants. Thus, when modelling the frequency of EBV mucosal shedding and viremia as a function of BAFF, we corrected for HIV-1 status, including this as a term in the GEE. Finally, GEE were used to compare the quantities of virus shed in mucosal samples in HIV-1 coinfected and HIV-1 uninfected participants assuming a Gaussian distribution. In all tests, two-sided p-values $\leq 0.05$ were considered statistically significant.

## Mathematical model simulations

Based on the reactions of the model, we applied the tau leaping algorithm to stochastically simulate the dynamics of each crypt [60]. With this algorithm, a small, constant-sized time step is taken, and the number of occurrences of each reaction is stochastically chosen following a Poisson or Multinomial distribution depending on the independence of the reaction. The population sizes of $T$, $I$ and $V$ are updated accordingly. For our simulations, we allowed the model to progress through time steps equal to 0.01 days, or 14.4 minutes. One long simulation is performed, which is then divided into 240 sections to represent the dynamics of each of the 240 crypts. Specifically, the simulation is run out to a time of

$$W_{\text{init}} + 240(L_i + W_{\text{crypt}}) \tag{7}$$

and crypt $c$'s dynamics are taken from the time interval

$$t \in \left[ W_{\text{init}} + (c-1)(L_i + W_{\text{crypt}}), W_{\text{init}} + cL_i \right] \tag{8}$$

where $W_{\text{init}}$ represents the time necessary to remove the effects of the initial conditions on the simulation, $L_i$ represents the duration of which participant $i$ had oral swabs taken, and $W_{\text{crypt}}$ represents the time necessary to make the dynamics of one crypt quasi-independent of the next. With immune cell decay ($\delta T$) acting as the slowest rate in the model ($\delta = 0.1/$(day-cell)), we let both $W_{\text{crypt}}$ and $W_{\text{init}}$ equal 120 days, so that if infection in one crypt occurred, only an expected $1/10^6$ of the responding immune cells would carry on to the next crypt's dynamics.

At the beginning of a simulation, initial conditions were set to describe an infection-free state where $V = 0$, $T = 0$ and $I = 0$.

Viral loads from each crypt are added together to get the model-predicted amount of virus seen in the saliva over time. As the qPCR threshold of detection was 150 copies/ml for the data used, whenever the total simulated viral load in the saliva dropped below 150 copies/ml, we set the output to zero.

## Model fitting using Approximate Bayesian Computation

We next fit parameters to daily quantitative oral EBV shedding data from our cohort participants. We used Approximate Bayesian Computation (ABC), where summary statistics of the data and model simulations are compared to determine which parameters allow the model to fit the data best. Fitting was performed separately for each set of participants' data. We used the R package EasyABC to execute sequential ABC, following Lenormand's algorithm [61]. Here, uniform priors for each parameter are set, and an initial $n$ number of simulations are run, each with a different set of parameters randomly chosen from the priors. The algorithm calculates summary statistics, chosen by the user, for each parameter set ($\hat{D}$) and compares how well they match with the summary statistics of the data ($D$) by calculating a distance measure, $\rho(D, \hat{D})$. The best-matching $\phi$ percent of these simulations are kept, with the parameters of the chosen simulations used to build new priors. This process repeats until the distance between the summary statistics of the data and simulations is minimized. We executed this algorithm for the data of each individual participant. We chose to capture the trends of the data using 5 summary statistics: the frequency of positive swabs, the median, maximum, and variance of detectable viral loads, and the number of peaks in viral loads, with a peak defined as when the directly preceding and following time points have lower viral loads. While more summary statistics could be considered desirable, too many can overwhelm the algorithm, leading to poor convergence to a posterior distribution. With these 5 summary statistics, the associated $\rho$ value for participant $i$ and parameter set $j$ ($\rho_{i,j}$) is defined as

$$\rho_{i,j} = \frac{1}{5} \sum_{k=1}^{5} \left| \frac{D_{i,k} - \hat{D}_{j,k}}{D_{i,k}} \right| \qquad (9)$$

where $D_{i,k}$ is the $k^{th}$ summary statistic for the data of participant $i$ and $\hat{D}_{j,k}$ is the $k^{th}$ summary statistic for parameter set $j$. Using the Lenormand algorithm, 1000 parameter sets that minimize $\rho_{i,j}$ were selected for each participant. During this fitting, we fixed all parameters except $b$ and $\theta$ as these are the two parameters that are likely most affected by HIV-1 infection. For each chosen parameter set, we also calculated the median number of actively infected crypts and the median amount of virus produced by an actively infected crypt so we could later compare how these traits differ between participant groups.

Since we lack information on immune cell presence in tonsillar crypts and can only fit the model to data on viral load, we had to censor simulations where the cytotoxic T cell level became unrealistically high. Whenever a parameter set led to a simulation where cytotoxic T cell count within a tonsillar crypt was greater than $1.475 \times 10^6$ cells (equivalent to $10^5$ cells/cm$^2$ which is the estimated maximum density in the genital tract during HSV-2 infection [62]), we prevented it from being selected by the ABC algorithm, ensuring only biologically relevant simulations were considered.

## Determining the posterior distribution of parameters and differences between participant groups

We combined the results of our ABC fitting algorithm to compare how the posterior distributions of parameters $b$ and $\theta$, the number of actively infected crypts, and the amount of virus produced vary between different participant groups.

As some parameter values selected by the ABC fitting algorithm fit the data better than others (i.e. have lower $\rho$ values), we approximated the posterior distributions of our parameter sets by performing importance sampling on the raw posterior distributions [63, 64]. To do this, we weighted each output parameter set by the reciprocal of its $\rho$ value. By weighting inversely to $\rho$, we assume our model is a correct representation of viral dynamics in the tonsils and put greater importance on those parameter sets that fit the data well.

To determine the posterior distributions of parameters in HIV-1 coinfected and uninfected groups ($X_A$ and $X_B$ respectively), the probability of each parameter set ($x_{(}i, j)$) serving in each posterior is set to

$$P(X_A = x_{i,j}) = P(i \in A) \frac{1}{\rho_{i,j}} \frac{1}{q_A} \tag{10}$$

$$P(X_B = x_{i,j}) = P(i \in B) \frac{1}{\rho_{i,j}} \frac{1}{q_B} \tag{11}$$

where $A$ and $B$ are the sets of indices of participants who are HIV-1 uninfected and coinfected, respectively, and we define the normalization factors

$$q_A = \sum_{\forall i \in A} \sum_{\forall j} \frac{1}{\rho_{i,j}} \quad \text{and} \quad q_B = \sum_{\forall i \in B} \sum_{\forall j} \frac{1}{\rho_{i,j}}. \tag{12}$$

Note that

$$\sum_{\forall i} \sum_{\forall j} P(X_A = x_{i,j}) = 1 \quad \text{and} \quad \sum_{\forall i} \sum_{\forall j} P(X_B = x_{i,j}) = 1. \tag{13}$$

We took $10^5$ draws from each distribution and plotted the resulting data to obtain graphical representations of the posterior parameter distributions for parameters $b$ and $\theta$, the number of actively infected crypts, and the virus produced, for HIV-1 coinfected and uninfected participants. The above process was repeated where instead of stratifying by HIV-1 status, participants were stratified by median EBV load in order to produce similar plots.

We also performed importance sampling on the raw posterior distributions for individual participants. Using these, we were able to calculate the mean parameter values for $b$ and $\theta$ for each participant. Means of parameters $b$ and $\theta$ in HIV-1 coinfected participants were then modelled as functions of the participants' CD4+ T cell count and HIV-1 RNA load. This was done using GLM, allowing for continuous adjustment for each 100 cell/mm$^3$ increase in CD4 + T cell count and each $\log_{10}$ increase in HIV-1 RNA. Parameters $b$ and $\theta$ were assumed to follow a Gaussian distribution. In these tests, two-sided p-values $\leq 0.05$ were considered statistically significant.

## Sensitivity analysis of model parameters

We performed two sensitivity analyses to evaluate how changes in parameter values affect the results of the model. First, to initially determine acceptable values for parameters, we performed a univariate analysis, starting with an initial set of parameters, varying one parameter

at a time, and running 100 simulations for each parameter set to gain a representation of how EBV viral dynamics behave. From this analysis, we selected parameter values for $\beta$, $f$, $\alpha$, $\delta$, $p$, and $c$ which would remain fixed throughout the ABC data fitting process while leaving the parameters of most interest, $b$ and $\theta$, free.

After completing ABC, we performed another univariate analysis to observe whether our choices for fixed parameter values were correct. By letting $b$ and $\theta$ equal values selected by the ABC algorithm and then individually varying the parameters that were fixed, we checked whether different values of our fixed parameters would have improved the model's fit. Results of these sensitivity analyses are found in S2 Text.

## Supporting information

**S1 Text. Analysis of Uganda cohort's genital swabs and plasma samples.**
(PDF)

**S2 Text. Mathematical model parameter estimation and goodness of fit.**
(PDF)

**S3 Text. Validation of the mathematical model using a North American cohort.**
(PDF)

## Acknowledgments

This work was enabled in part by support provided by WestGrid and Compute Canada.

## Author Contributions

**Conceptualization:** Catherine M. Byrne, Christine Johnston, Jackson Orem, Fred Okuku, Anna Wald, Lawrence Corey, Corey Casper, Daniel Coombs, Soren Gantt.

**Data curation:** Catherine M. Byrne, Christine Johnston, Meei-Li Huang, Anna Wald.

**Formal analysis:** Catherine M. Byrne, Christine Johnston, Meei-Li Huang, Daniel Coombs.

**Funding acquisition:** Catherine M. Byrne, Christine Johnston, Anna Wald, Lawrence Corey, Corey Casper, Daniel Coombs, Soren Gantt.

**Investigation:** Catherine M. Byrne, Christine Johnston, Jackson Orem, Fred Okuku, Meei-Li Huang, Habibur Rahman, Anna Wald, Lawrence Corey, Corey Casper, Daniel Coombs, Soren Gantt.

**Methodology:** Catherine M. Byrne, Christine Johnston, Jackson Orem, Fred Okuku, Meei-Li Huang, Habibur Rahman, Anna Wald, Lawrence Corey, Joshua T. Schiffer, Corey Casper, Daniel Coombs, Soren Gantt.

**Project administration:** Catherine M. Byrne, Christine Johnston, Jackson Orem, Fred Okuku, Anna Wald, Lawrence Corey, Corey Casper, Daniel Coombs, Soren Gantt.

**Resources:** Catherine M. Byrne, Christine Johnston, Anna Wald, Lawrence Corey, Daniel Coombs, Soren Gantt.

**Software:** Catherine M. Byrne, Daniel Coombs.

**Supervision:** Christine Johnston, Anna Wald, Joshua T. Schiffer, Daniel Coombs, Soren Gantt.

**Validation:** Catherine M. Byrne, Christine Johnston, Soren Gantt.

**Visualization:** Catherine M. Byrne.

**Writing – original draft:** Catherine M. Byrne, Christine Johnston, Daniel Coombs, Soren Gantt.

**Writing – review & editing:** Catherine M. Byrne, Christine Johnston, Meei-Li Huang, Anna Wald, Lawrence Corey, Joshua T. Schiffer, Corey Casper, Daniel Coombs, Soren Gantt.

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
