## [Decision Letter · Decision Letter 0]

18 Dec 2020

Dear Ms. Byrne,

Thank you very much for submitting your manuscript "Examining the dynamics of Epstein-Barr virus shedding in the tonsils and the impact of HIV-1 coinfection on daily saliva viral loads" for consideration at PLOS Computational Biology.

As with all papers reviewed by the journal, your manuscript was reviewed by members of the editorial board and by several independent reviewers. In light of the reviews (below this email), we would like to invite the resubmission of a significantly-revised version that takes into account the reviewers' comments.

We cannot make any decision about publication until we have seen the revised manuscript and your response to the reviewers' comments. Your revised manuscript is also likely to be sent to reviewers for further evaluation.

Sincerely,

Andrew J. Yates

Associate Editor

PLOS Computational Biology

Rob De Boer

Deputy Editor

PLOS Computational Biology

Reviewer's Responses to Questions

**Comments to the Authors:**

Reviewer #1: This study investigates the difference in virus shedding and daily saliva viral loads of EBV between HIV-1 co-infected and HIV-1 uninfected individuals. The authors used a large dataset (85 individuals) and find that HIV-1 co-infection leads to greater and more frequent oral EBV shedding. The authors have developed a new, stochastic and mechanistic model of EBV infection to study the mechanism of the greater EBV shedding of HIV-1 co-infected individuals. This model is the first one to describe longitudinal EBV shedding data. It estimates two of the parameters of this model from data and fixes all other parameters based on a sensitivity analysis and literature. These two parameters (b and theta), the reactivation of latently infected b cells and the strength of the cellular immune response, are estimated for each individual separately. The authors then compare the distribution of these parameters for HIV-1 co-infected and HIV-1 uninfected individuals. Their main conclusion is that greater oral EBV shedding with HIV-1 coinfection is due to both increased B cell reactivation and weaker cellular immune response.

I find this study strong in both data and methodology. All model assumptions are described and argued for in the results. For instance, the authors clearly explain why the model contains 240 different tonsils and why it should be stochastic. The dataset used by the authors is large and the main findings from the study are even validated with a second dataset. I think that this manuscript is suitable for publication in PLOS computational biology. I only have two minor comments.

Minor comments

I understand that the authors have focused on testing whether either b or theta or both can explain the differences in EBV shedding between HIV-1 uninfected and HIV-1 co-infected individuals. They also clearly describe this in the second paragraph on page 12. The authors have good biological arguments on why differences in b and theta are expected. Also, I agree that fitting all parameters is not feasible. However, I am curious which other parameters could theoretically explain the difference as well, and why these are biologically unlikely.

The readability of the sentence ‘as our estimates … to be justified’ on lines 134-136 could be improved.

Reviewer #2: I enjoyed reading this paper using mathematical models and a powerful

data set to analyze the interaction between HIV and EBV infections. The

stochastic model based on individual crypts builds nicely on previous

work and the biology of this herpesvirus, and gave me a lot to think

about. I do have some concerns with the organization, interpretation and

presentation that I hope will help improve the paper.

A. Even though I don't think this affects the results, the writing

treats HIV as causal of differences in EBV dynamics. Because this is

observational, there is no way to distinguish this from intrinsic

differences in patients. That is, those more likely to acquire HIV

might also be those more likely to have, for example, multiple

coinfecting strains of EBV. Please check the language to be careful

about this issue of interpretation. This issue arises also on page 9,

where CD4+ T cell counts etc are correlated with EBV detection, rather

than impacting it.

As a related point, I found it confusing not to have information about

any patient covariates that could capture some of this variation.

B. The paper seemed a bit out of order to me (with some of my thoughts

included in my comments on the supplement below). I think that the

results on the distributions of b and theta should be first, explaining

what the values mean for the dynamics more fully. One thing that would

help me would be an explanation of why EBV infections in fact die out in

a crypt. Could this be linked with an epidemiological model and thus

have a calculation of R0 that would help with interpretation? This

could even precede the results on b and theta from patients, with the

comparison of HIV positive and negative patients last. Finally, I think

many more of the results from the Seattle cohort should appear in the

main paper.

C. I have a few questions about the choice of statistics to present and

to use in the ABC algorithm. Page 4 discusses shedding episodes, but I

couldn't find a precise definition. More importantly, why isn't a

statistic like this used in the ABC algorithm, because it should have

much more resolution than the ones shown for capturing the temporal

patterns in the data. For example, are the data autocorrelated within

patients, and would this provide a more robust statistic than run

length, which is very sensitive to false negatives? I don't think

Figure 2A and 2B add much information. And perhaps violin plots might

be a good choice for the data in 2C and 2D.

Along these lines, I found myself wondering whether it is possible to

detect the number of infected crypts from the data, given that they come

in small numbers. I've seen methods for doing this in cell physiology,

where people can tell how many calcium channels are open. One paper is

https://doi.org/10.1073/pnas.96.24.13750

although I'm sure there are many more recent ones, and I'm pretty sure

people do similar things in neuroscience.

D. I think the use of tables and figures could be improved. I've noted

a couple of places where the text has a lot of numbers that are

difficult to read, or where the statistics are difficult to find. On

page 9, the information from Table 1 is largely repeated in the text.

I'm not sure whether using cumulative distributions, or heatmaps would

be a better way, but I found Figures 4 and 5 to be pretty weak in

conveying information. Perhaps cdf's for Figure 4 and heatmaps with b

and theta on the axes and color indicating HIV status and viral load

would expose the patterns more clearly.

E. The paper frequently emphasizes that the models are "new" or "novel"

with many appearances of the word "unique". None of this is needed

because this work speaks for itself, in my view.

MORE MINOR POINTS

Page 1, last line: Could just say "we developed a stochastic...".

This is one of many places where the word "data" is treated as singular,

although it is technically plural.

Abstract, last line: I'm not an expert, but I'd be a bit wary of

recommending B cell reactivation as a therapeutic target. There could

be a lot of side effects.

Page 3, line 29: I don't think "large" is needed here.

Page 3, line 55: I think "determined" is too strong, perhaps

"estimated".

Page 3, line 59: The introduction should give the number of patients in

the Seattle cohort.

Page 3, line 60: The final sentence is rather vague.

Page 4, line 65: This paragraph is loaded with a lot of numbers that are

hard to read. There should be a way to incorporate this information

into the figure and the figure legends to make the text easier to

follow.

Page 4, figure 2 legend: "Sustained viral shedding" is unclear here.

Page 5, line 131: Are there latently infected cells in the Waldeyer's

ring?

Page 5, line 144: It is probably just my ignorance, but are there really

tissue-resident T cells?

Page 7, line 205: "show" is a bit strong. This is a model result.

Page 8, Figure 4 legend: The last sentence states what is basically a

tautology; it has to be one or the other. The key is partitioning into

these two components to say which is more important.

Page 8, line 242: No need to start a sentence with "We also note that.."

Page 10: Why would HIV-1 RNA have a stronger association?

Page 10, line 281: "further validating results" seems strong to me. And

this is another paragraph that repeats the information in the Table and

is hard to follow.

Page 11, line 319: Not sure that "worse" is clearly defined here.

Page 11, line 327: The word "alone" is not needed.

Page 11, line 330: Maybe I missed it, but there is some confusion

between showing associations in patients and showing associations with

estimated parameters. I'd like to see a description of covariation of

measurements in patients in the main paper to motivate the association

with estimated parameters. This could also include information on

patient covariates if available.

Page 14, line 464: How long is the transient and what are the initial

conditions?

Page 15, line 507: How often did this happen, and why was this threshold

chosen?

SUPPLEMENT

The rather long-named section "Association between participants HIV

infection severity, CD4+ T cell characteristics, and B cell activation

measures and their EBV loads in genital swabs and plasma samples"

belongs in the main text, in my view, although it needs to be tightened

up substantially. Some clarification of the interpretation of genital vs

plasma swabs is needed. It would really help to incorporate the statistics

into the figures, which I hope makes the two tables redundant and

unnecessary.

Be much more careful about casual language, for example on Page 2,

"we saw that increases in BAFF led to increases". As a separate issue,

I do not think that signals this weak should be discussed. In the next

sentence, a comparison is made between plasma and genital swabs based on

p-values, as far as I can understand. This is not valid, and needs to

looked at through a single model with an interaction effect.

The section "Basic model analysis" on page 4 seems to be about parameter

estimates and is perhaps misnamed. The paragraph on this page is very

hard to follow, and should be linked more clearly with the figure. It

seems there are two criteria: estimates from the literature, and some

set of targets for what is reasonable. It would help to have those

target criteria laid out in advance so that the figure can be

interpreted. Finally, some further clarification about the comparison

with the estimates from the literature would help. I'm not sure what

"which generally agreed with published estimates" means for beta.

I have no problem with the way this was done, but would it be possible

to set up the calibration targets and do a multivariate search for

parameters that hit those targets?The section "Mathematical model fits clinical data well and simulates

oral shedding data with high fidelity" on page 7 repeats a fair amount

of material from the main text.

The sensitivity analysis is a bit confusing. I understand that a

comprehensive analysis is impossible, and likely not very interesting.

Fixing b and theta, and then choosing other values of the fixed

parameters is hard for me to understand. Is it impossible to redo the

analysis, meaning finding b and theta, for the 12 values shown in Figure

S5? One could then ask whether the main findings hold up. If that is

too computationally expensive, maybe just do a couple of cases where the

parameter estimates are most uncertain or most inconsistent with

empirical values.

The section on the Seattle study could also be focused and included in

the main text, particularly if the authors can find a more efficient way

to present the data graphically. The writing here is hard to follow,

with too many numbers included in the text.

The final paragraph is not appropriate for a supplement, which should be

focused on methods and detailed results, not on interpretation.

Reviewer #3: The mathematical model is stochastic, with three populations and seven

parameters. Most parameter values are simply guessed (and fixed) while

two are fitted to the data. The model and its behaviour is not fully

described, even in the supplementary material, so I have to deduce

part of what follows: the three variables change by one unit at a

time, according to the Gillespie algorithm. This is not important in

the case of $V$ because one infected cell produces $p=10^4$ virions

per day (I think the unit ml$^{-1}$ is added in error in table

S3). Thus if $I$ were constant we would expect a viral load of

$\\frac{p}cI$. The population $T$ is large, because it is assumed that

a minimum of $\\alpha=200$ cells per crypt are present. The key to the

modelling, I think, is that $I$ is often equal to $0$ and when not

zero, it is a small integer for some time. Thus the authors'

unconventional numerical method of not actually simulating 240

independent processes, but performing one long run and dividing the

timeseries up, produces acceptable results. If all this is correct, please

say so. If not, please explain the dynamics of the model. Show

some sample trajectories.

**Have all data underlying the figures and results presented in the manuscript been provided?**

Reviewer #1: **No: **I could not get acces to the provided data source: doi:10.5061/dryad.w6m905qkh. However, this might only be public upon publication.

Reviewer #2: Yes

Reviewer #3: Yes

PLOS authors have the option to publish the peer review history of their article (what does this mean?). If published, this will include your full peer review and any attached files.

Reviewer #1: No

Reviewer #2: No

Reviewer #3: **Yes: **Grant Lythe
---

## [Decision Letter · Decision Letter 1]

12 Apr 2021

Dear Ms. Byrne,

Thank you very much for submitting your manuscript "Examining the dynamics of Epstein-Barr virus shedding in the tonsils and the impact of HIV-1 coinfection on daily saliva viral loads" for consideration at PLOS Computational Biology. As with all papers reviewed by the journal, your manuscript was reviewed by members of the editorial board and by several independent reviewers. The reviewers appreciated the attention to an important topic. Based on the reviews, we are likely to accept this manuscript for publication, providing that you modify the manuscript according to the review recommendations.

Thank you for your patience while we gathered these reviews. You will see a few further comments and corrections requested by Reviewer #2 - please pay careful attention to these. We will turn this around as quickly as possible once we receive your revision.

Sincerely,

Andrew J. Yates

Associate Editor

PLOS Computational Biology

Rob De Boer

Deputy Editor

PLOS Computational Biology

[LINK]

Thank you for your patience while we gathered these reviews. You will see a few further comments and corrections requested by Reviewer #2 - please pay careful attention to these. We will turn this around as quickly as possible once we receive your revision.

Reviewer's Responses to Questions

**Comments to the Authors:**

Reviewer #1: My comments have been addressed satisfactorily and I have no additional comments. I recommend accepting this paper for publication.

Reviewer #2: Thanks for the careful revisions of this paper, which presents an

elegant example of mathematical modeling in action. I have just a few

very minor editorial comments, and one question about the model.

My question about the model is the T + alpha in equation (3). If alpha

represents tissue-resident T cells, why aren't they just constant,

making alpha appear only in equation (2)?

Minor points:

Summary: "both detectable and high quantities" sounds odd, but I'm not

sure how to fix it.

Line 31: "These data capture"

Line 49: Maybe cut "uniquely"

Lines 77-78: I'm not sure what "cannot predict" means here. Is there a

statistical test?

Lines 93-95: Sounds like a regression is being run, ideally with an

interaction term. It would be nice to show the results and make clear

what the p-value corresponds to.

Line 151: "for at" has an extra word.

Figure 5: My figures are a bit fuzzy, but the arrows are hard to see.

Line 257: Extra period.

Line 282: "chose to" seems unnecessary.

Figure 7 legend, line -2: "than a" maybe should be "compared with"

Line 293: Maybe I missed this, but is the model prediction of BAFF level

made explicit?

Line 328: "where" should be "were".

Line 366: How about "is" instead of "was uniquely"?

**Have the authors made all data and (if applicable) computational code underlying the findings in their manuscript fully available?**

Reviewer #1: Yes

Reviewer #2: Yes

PLOS authors have the option to publish the peer review history of their article (what does this mean?). If published, this will include your full peer review and any attached files.

Reviewer #1: No

Reviewer #2: No

Figure Files:

Data Requirements:

Reproducibility:

References:

---

## [Editor Report · Decision Letter 2]

12 May 2021

Dear Ms. Byrne,

We are pleased to inform you that your manuscript 'Examining the dynamics of Epstein-Barr virus shedding in the tonsils and the impact of HIV-1 coinfection on daily saliva viral loads' has been provisionally accepted for publication in PLOS Computational Biology.

Best regards,

Andrew J. Yates

Associate Editor

PLOS Computational Biology

Rob De Boer

Deputy Editor

PLOS Computational Biology

---

## [Editor Report · Acceptance letter]

16 Jun 2021

PCOMPBIOL-D-20-01942R2 

Examining the dynamics of Epstein-Barr virus shedding in the tonsils and the impact of HIV-1 coinfection on daily saliva viral loads

Dear Dr Byrne,

I am pleased to inform you that your manuscript has been formally accepted for publication in PLOS Computational Biology. Your manuscript is now with our production department and you will be notified of the publication date in due course.

With kind regards,

Agota Szep
